# Haloperidol Attenuates Lung Endothelial Cell Permeability In Vitro and In Vivo

**DOI:** 10.3390/cells10092186

**Published:** 2021-08-25

**Authors:** Marco A. Colamonici, Yulia Epshtein, Weiguo Chen, Jeffrey R. Jacobson

**Affiliations:** 1Department of Medicine, Division of Pulmonary, Critical Care, Sleep and Allergy, University of Illinois Chicago, Chicago, IL 60607, USA; macolamonici@gmail.com (M.A.C.); yuliaa@uic.edu (Y.E.); weiguo@uic.edu (W.C.); 2Room 920N (MC719), 840 S. Wood St., Chicago, IL 60612, USA

**Keywords:** haloperidol, claudin-5, endothelial cells, vascular permeability, acute lung injury

## Abstract

We previously reported that claudin-5, a tight junctional protein, mediates lung vascular permeability in a murine model of acute lung injury (ALI) induced by lipopolysaccharide (LPS). Recently, it has been reported that haloperidol, an antipsychotic medication, dose-dependently increases expression of claudin-5 in vitro and in vivo, in brain endothelium. Notably, claudin-5 is highly expressed in both brain and lung tissues. However, the effects of haloperidol on EC barrier function are unknown. We hypothesized that haloperidol increases lung EC claudin-5 expression and attenuates agonist-induced lung EC barrier disruption. Human pulmonary artery ECs were pretreated with haloperidol at variable concentrations (0.1–10 μM) for 24 h. Cell lysates were subjected to Western blotting for claudin-5, in addition to occludin and zona occludens-1 (ZO-1), two other tight junctional proteins. To assess effects on barrier function, EC monolayers were pretreated for 24 h with haloperidol (10 µM) or vehicle prior to treatment with thrombin (1 U/mL), with measurements of transendothelial electrical resistance (TER) recorded as a real-time assessment of barrier integrity. In separate experiments, EC monolayers grown in Transwell inserts were pretreated with haloperidol (10 µM) prior to stimulation with thrombin (1 U/mL, 1 h) and measurement of FITC-dextran flux. Haloperidol significantly increased claudin-5, occludin, and ZO-1 expression levels. Measurements of TER and FITC-dextran Transwell flux confirmed a significant attenuation of thrombin-induced barrier disruption associated with haloperidol treatment. Finally, mice pretreated with haloperidol (4 mg/kg, IP) prior to the intratracheal administration of LPS (1.25 mg/kg, 16 h) had increased lung claudin-5 expression with decreased lung injury as assessed by bronchoalveolar lavage (BAL) fluid protein content, total cell counts, and inflammatory cytokines, in addition to lung histology. Our data confirm that haloperidol results in increased claudin-5 expression levels and demonstrates lung vascular-protective effects both in vitro and in vivo in a murine ALI model. These findings suggest that haloperidol may represent a novel therapy for the prevention or treatment of ALI and warrants further investigation in this context.

## 1. Introduction

Acute lung injury (ALI), which manifests clinically as acute respiratory distress syndrome (ARDS), is a significant cause of morbidity and mortality in intensive care units globally, with mortality rates as high as 46% for patients with severe ARDS [1]. Most recently, in the context of the coronavirus infectious disease 2019 (COVID-19) pandemic caused by severe acute respiratory syndrome coronavirus 2 (SARS-CoV-2), an estimated one-third of hospitalized patients developed ARDS [2]. Despite the clinical magnitude of the problem, no effective targeted ARDS therapies exist and treatment remains largely supportive, including lung-protective low tidal volume mechanical ventilation [3] and the use of interval prone positioning to optimize oxygenation [4].

ALI is characterized by an extensive inflammatory cascade that ultimately impairs gas exchange and lung function. A cardinal feature of ALI is increased lung endothelial cell (EC) permeability, resulting in increased transit of cells, protein-rich fluid, and inflammatory signaling molecules into the surrounding lung parenchyma and alveoli [5]. Direct consequences of this dysregulated inflammatory process are tissue hypoxia from pulmonary edema and the need for mechanical ventilatory support. Thus, strategies aimed at mitigating lung vascular permeability offer promise as novel and effective ALI therapies. One potential target in this regard is the EC tight junctional protein, claudin-5.

EC tight junctions are composed of transmembrane proteins, including claudins, occludins, and junctional adhesion molecules, that associate with cytoplasmic proteins, including zonula occludens (ZO). The importance of tight junctions in regulating EC permeability is evidenced by increased permeability of the blood–brain barrier in mice deficient in claudin-5 [6], which is predominantly expressed in ECs [7]. Moreover, we previously reported claudin-5 is an important mediator of ALI protection by simvastatin in a murine model [8]. Subsequent reports have further supported a significant functional role for claudin-5 in lung EC barrier regulation [9]. Notably, a recent study identified increased expression of claudin-5 in brain ECs induced by the antipsychotic medication, haloperidol [10]. Thus, we investigated the effect of haloperidol on lung EC claudin-5 expression and its role as a potential mediator of EC permeability relevant to ALI.

## 2. Materials and Methods

### 2.1. Antibodies and Reagents

Antibodies against claudin-5, ZO-1, occludin (Santa Cruz Biotechnology, Santa Cruz, CA, USA), β-actin (Sigma, St. Louis, MO, USA), horseradish peroxidase-conjugated anti-mouse and anti-rabbit secondary antibodies (Cell Signaling, Danver, MA, USA) were purchased from the indicated vendors. Other used reagents, including claudin-5 siRNA (Dharmacon, Lafayette, CO, USA), haloperidol and thrombin (Sigma) were also commercially purchased. A dose of 10 µM haloperidol was used for in vitro studies after preliminary experiments confirmed a dose-response effect with the most prominent changes in protein expression evident with this dosing. Specific experimental conditions are otherwise detailed in the text.

### 2.2. Endothelial Cell Culture

Human pulmonary artery endothelial cells (ECs) were cultured in essential growth medium (EGM-2) containing 10% fetal bovine serum (Clonetics, Walkersville, MD, USA). Cells were placed in an incubator at 37 °C, 5% CO_2_, and 95% humidity to achieve contact-inhibited monolayers.

### 2.3. Immunoblotting

Total proteins were extracted using NP-40 lysis buffer (50 mM TrisHCl pH 7.4, 150 mM NaCl, 1% NP-40, and 5 mM ethylenediaminetetraacetic acid) supplemented with 40 mM sodium fluoride, 0.1 M sodium orthovanadate, 0.2 mM phenylmethylsulfonyl fluoride, 10 mM N’ ethyl malamide, and protease and phosphatase inhibitor cocktail (Calbiochem, San Diego, CA, USA). Lung homogenates and cell lysates were briefly sonicated and subjected to cycles of thawing and freezing on dry ice. The protein concentrations were measured using a bicinchoninic acid protein assay kit (Pierce, Rockford, IL, USA). Western blotting was performed using standard protocols and band densities were determined using ImageJ. 

### 2.4. Transendothelial Electrical Resistance (TER) Measurement

ECs were plated in polycarbonate wells containing evaporated gold microelectrodes to measure TER, which evaluates real-time changes in cell morphology, attachment, and locomotion using an electric cell-substrate impedance system (ECIS, Applied Biophysics, Troy, NY, USA), as we previously reported [8]. Cells were grown to confluence in EBM-2 containing 2% serum prior to treatment with haloperidol (10 µM) for 24 h followed by the administration of thrombin (1 U/mL), which is known to disrupt barrier integrity. TER values from each microelectrode were pooled at discrete time points and plotted versus time as the mean ± SEM. For siRNA experiments, cells were first transfected with siRNA (100 mM, 3 d) prior to treatment with haloperidol, followed by thrombin stimulation, as described above.

### 2.5. Transwell Permeability Assay

A commercially available kit (Millipore, Billerica, CA, USA) was used to measure EC monolayer permeability to high molecular weight proteins based on the Transwell model our laboratory previously described [8]. In these experiments, EC monolaryers were pretreated with haloperidol (10 µM) for 24 h, after which 100 µL FITC-dextran (2000 kDa) was added to cells at the same time as thrombin (1 U/mL) and incubated for 1 h. The Transwell insert was then removed and 100 µL medium collected. Fluorescent density was analyzed on a Titertek Fluoroskan II Microplate Fluorometer (Diversified Equipment, Lorton, VA, USA) at excitation and emission wavelengths of 485 and 530 nm, respectively.

### 2.6. Murine ALI Model

All experiments and animal care procedures were approved by the University of Illinois at Chicago Animal Care and Use Committee. C57/Bl6 mice (Jackson Labs, Bar Harbor, ME, USA), 8–12 weeks old, were administered haloperidol (4 mg/kg, IP) or vehicle 24 h prior to repeat treatment with haloperidol or vehicle and intratracheal LPS (1.25 mg/kg body weight). The next day, 24 h after LPS, the mice were sacrificed and bronchoalveolar lavage (BAL) fluid was collected and assessed for total protein, in addition to total cell counts and inflammatory cytokines, as previously described [11]. Lungs from select animals were harvested and used for either preparation of homogenates subjected to Western blotting for claudin-5 or for histology.

### 2.7. Statistics

Student’s *t*-test was used to compare the means of data from two experimental groups and significant differences (*p* < 0.05) among multiple group comparisons were confirmed by one-way ANOVA followed by Tukey’s range test. Results are expressed as means ± SE.

## 3. Results

### 3.1. Effect of Haloperidol on Lung EC Expression of Tight Junctional Proteins

In our initial experiments, human pulmonary artery ECs were treated with haloperidol at variable dosing (0.1, 1.0, or 10 µM) for 24 h, and cell lysates were collected and subjected to Western blotting for claudin-5 expression (Figure 1A). These experiments confirmed a significant increase in claudin-5 levels at all three dosings. Importantly, RT-PCR experiments did not detect a significant change in claudin-5 mRNA levels (data not shown). These findings are consistent with results from similar studies utilizing brain ECs [10]. Furthermore, in separate experiments with the same conditions, treatment with haloperidol was also associated with a significant increase in expression levels of ZO-1 and occludin (Figure 1B).

### 3.2. Effect of Haloperidol on Lung EC Barrier Function

To investigate the effects of haloperidol on EC barrier regulation, we initially measured transendothelial electrical resistance (TER) in ECs pretreated with haloperidol prior to thrombin stimulation. These studies confirmed that maximal barrier disruption post-thrombin was significantly attenuated by haloperidol (Figure 2A).

In separate experiments, human pulmonary artery ECs were grown to confluence in Transwell inserts prior to treatment with haloperidol. FITC-dextran (2000 kDa) Transwell flux across monolayers after thrombin was then assessed by collecting the media below each insert and calculating fluorescence density as a direct measurement of monolayer permeability. Consistent with the TER studies, a significant attenuation of thrombin-induced FITC-dextran flux was observed with haloperidol treatment (Figure 2B).

### 3.3. Role of Claudin-5 in Lung EC Barrier Protection by Haloperidol

To study the role of claudin-5 in the lung EC barrier-protective effects of haloperidol, TER experiments were repeated in cells transfected with silencing RNA specific for claudin-5 or non-specific RNA (Figure 3). Western blotting confirmed significant knockdown of claudin-5 by siRNA and TER measurements confirmed a significant attenuation of thrombin-induced barrier disruption associated with claudin-5 knockdown.

### 3.4. Effect of Haloperidol on Lung Claudin-5 Expression In Vivo

Finally, we employed a murine model of LPS-induced ALI to investigate the potential protective effects of haloperidol in this context. Mice were treated with haloperidol or vehicle for 24 h prior to a repeat treatment with either haloperidol or vehicle and the administration of intratracheal LPS. In select animals, lungs were harvested 24 h after LPS and homogenates were used for Western blotting for claudin-5 (Figure 4). Haloperidol pretreatment was associated with a significant increase in claudin-5 expression levels relative to controls. Western blots of whole lung homogenates were also probed for both ZO-1 and occludin but there was no appreciable change in expression levels of either protein (data not shown).

### 3.5. Effect of Haloperidol on Murine ALI: BAL Fluid Analysis

In separate animals, BAL fluid was collected 24 h after LPS for analysis of total protein content and cell counts (Figure 5), in addition to levels of inflammatory cytokines including IL-6, KC, MCP-1, and IL-1β (Figure 6). Of note, in an effort to best utilize animals for these studies, animal numbers were intentionally distributed in favor of LPS-treated conditions (*n* = 5/LPS treatment groups with *n* = 3/haloperidol alone and *n* = 2/control groups). These studies confirmed a significant attenuation of LPS-induced increases in BAL total protein content, total cell counts, and each of the measured inflammatory cytokines consistent with reduced lung injury.

### 3.6. Effect of Haloperidol on Murine ALI: Lung Histology

Separately, lungs were harvested from animals using the same experimental conditions and utilized for histologic evaluation. LPS alone was associated with a prominent infiltration of neutrophils into the lung parenchyma with evidence of interstitial edema. These changes were markedly reduced in LPS-treated animals that also received haloperidol (Figure 7).

## 4. Discussion

Haloperidol is an antipsychotic medication previously found to induce increased claudin-5 expression in brain ECs. We have now extended these findings and confirmed similar effects in lung ECs. In addition, as we previously reported regarding lung vascular-protective effects mediated by claudin-5, we investigated the barrier protective properties of haloperidol and observed lung EC barrier protection by haloperidol in vitro mediated by claudin-5. Finally, utilizing a murine model of ALI, we also confirmed increased lung claudin-5 and protective effects associated with haloperidol treatment in vivo.

These results are particularly intriguing given the extensive experience with the clinical use of haloperidol in critically ill patients, most often in an effort to manage or prevent delirium. Moreover, extrapolating from the literature in this area, there is strong evidence that this drug is safe and well-tolerated when given as a recurrent and prolonged treatment. For example, the REDUCE study randomized over 1700 ICU patients to receive intravenous haloperidol, 1 or 2 mg, three times daily, or placebo, in patients at high risk for delirium, defined by an anticipated ICU stay of at least 2 days [12]. Although the authors reported no mortality difference at 28 days associated with haloperidol treatment, the primary endpoint, they also reported no significant differences between the treatment and placebo groups with respect to all 15 secondary clinical endpoints, including ICU and hospital lengths of stay, and there was no difference in the incidence of adverse events reported. Although haloperidol has never been investigated specifically as a treatment for patients with or at risk for ALI, this study and several others support the feasibility of its use in this context [13,14,15].

Importantly, our animal studies relied on a relatively high dosing of haloperidol (4 mg/kg) that would clearly exceed the level considered safe in humans. This dosing in mice, however, was specifically chosen for our experiments because it has been found to correlate with serum concentrations typically achieved in humans [16]. Moreover, safety concerns regarding the use of haloperidol as treatment for ALI are mitigated, in part, by the fact that any treatment would be expected to be relatively short-term and therefore less likely to be associated with adverse effects seen with chronic treatment.

It is interesting to note that another antipsychotic medication, lithium chloride, has also been reported to increase EC claudin-5 expression levels [10]. Our current results suggest that similar studies of lithium chloride on lung EC permeability may also be warranted. Admittedly, little is known about the safety of lithium chloride use in ICU patients, and the most relevant literature in this area is largely focused instead on the management of patients with lithium toxicity [17,18]. Nonetheless, lithium levels are able to be rapidly measured and lithium toxicity is fully reversible with fluids and dialysis. Thus, the feasibility of therapeutic lithium use in the ICU should not be discounted.

Our findings in support of our hypothesis that haloperidol increases lung EC claudin-5 expression and augments EC barrier integrity relevant to ALI should also support the investigation of other, unrelated mediators of claudin-5 regulation in this context. One example of this is 17β-estradiol (E2), a natural endogenous estrogen that has been found to both upregulate claudin-5 in ECs and to augment EC barrier function as measured by TER [19]. The relevance of these effects is suggested by evidence of reduced lung inflammation associated with E2 administration in a rat model of ALI induced by intestinal ischemia and reperfusion [20]. Separately, in an LPS-ALI model, female ovariectomized mice were found to have significantly increased lung injury compared to female control mice, with no difference detected in ovariectomized mice who received estradiol compared to control females [21]. Multiple other reports have confirmed similar findings with respect to the protective effects of estrogen on ALI [22,23,24] and these observations are further supported by observed gender differences associated with ALI mortality [25]. The therapeutic potential of estrogen to treat ALI and the role of claudin-5 upregulation in this context are areas that warrant further study.

We recognize that one limitation in our ability to interpret our results is that our findings may not be solely due to increased claudin-5 associated with haloperidol treatment. We also observed the upregulation of two other tight junctional proteins, occludin and ZO-1, may also be an important contributor to these effects. Furthermore, effects of these drugs beyond those we investigated could also be involved. For example, haloperidol induces activation of glycogen synthase kinase-3 beta (GSK3β) [26], a serine/threonine kinase, and inactivates AMPK (AMP-activated protein kinase) [27]. These two signaling events have been shown to accentuate inflammation in ALI models [28]. Further study is needed to precisely define the mechanisms by which haloperidol affects EC barrier protection.

Our results support a significant lung vascular-protective effect of haloperidol. The translational relevance of these findings is evident because haloperidol is a drug that is both already in wide use in the ICU and has a relatively favorable safety profile. Further investigation into the therapeutic potential of haloperidol specifically for patients with or at risk for ALI is needed, and future study of other agents that are able to upregulate EC claudin-5 should also include their potential as novel ALI therapeutics.

## Figures and Tables

**Figure 1 cells-10-02186-f001:**
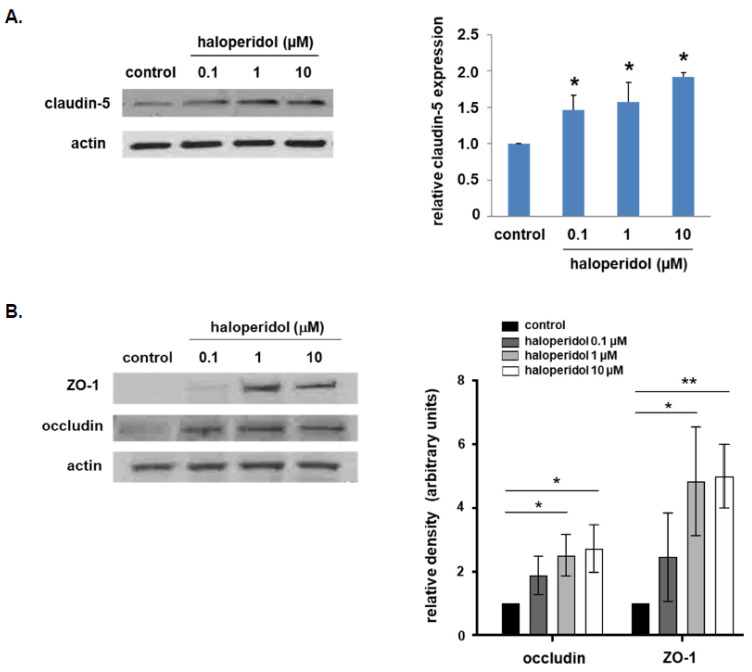
Claudin-5, ZO-1, and occludin expression is increased in lung ECs after haloperidol treatment. Human pulmonary artery ECs were treated with haloperidol (0.1, 1.0, or 10 µM) for 24 h and whole cell lysates were then collected and subjected to Western blotting for (**A**) claudin-5 or (**B**) ZO-1 or occludin with untreated cells used as controls. Expression of all three tight junctional proteins was significantly increased in response to haloperidol with an appreciable dose-dependent response (*n* = 3, * *p* < 0.05 and ** *p* < 0.01 compared to respective untreated controls; representative blots shown).

**Figure 2 cells-10-02186-f002:**
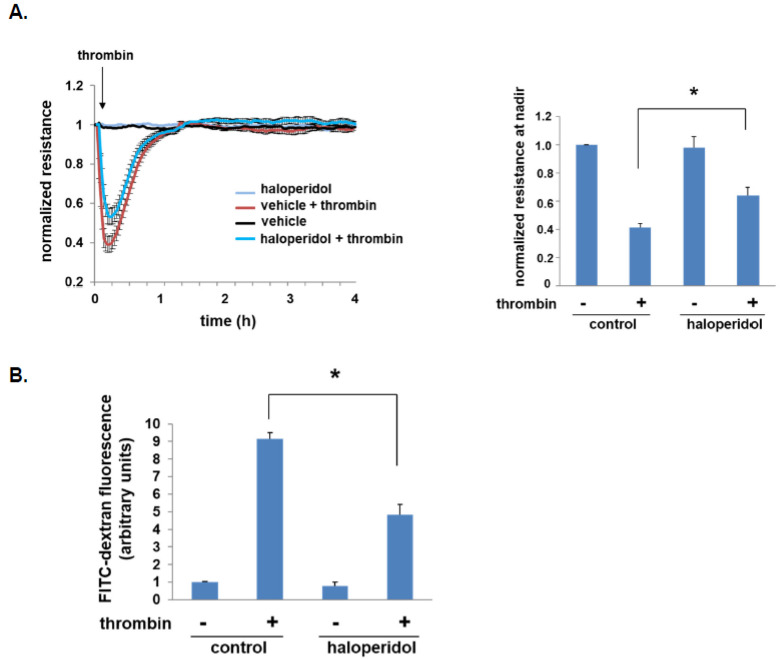
Haloperidol augments EC barrier function. (**A**) Human pulmonary artery ECs were grown to confluence overlying gold-plated microelectrodes for measurements of transendothelial electrical resistance. Cells were pretreated with haloperidol (10 µM) for 24 h prior to thrombin stimulation (1 U/mL). Real-time measurements of normalized resistance were recorded (left panel), and the nadir for each condition was used for the bar graph (right panel; *n* = 3/condition, * *p* < 0.05). (**B**) In separate experiments, human pulmonary artery ECs were grown to confluence in Transwell inserts. Cell monolayers were then treated with haloperidol (10 µM) for 24 h prior to the simultaneous administration of FITC-dextran (2000 kDa) to the media in each insert and treatment with thrombin (1 U/mL). After 1 h, the media underneath each insert was collected and fluorescence measured (*n* = 3/condition, * *p* < 0.05).

**Figure 3 cells-10-02186-f003:**
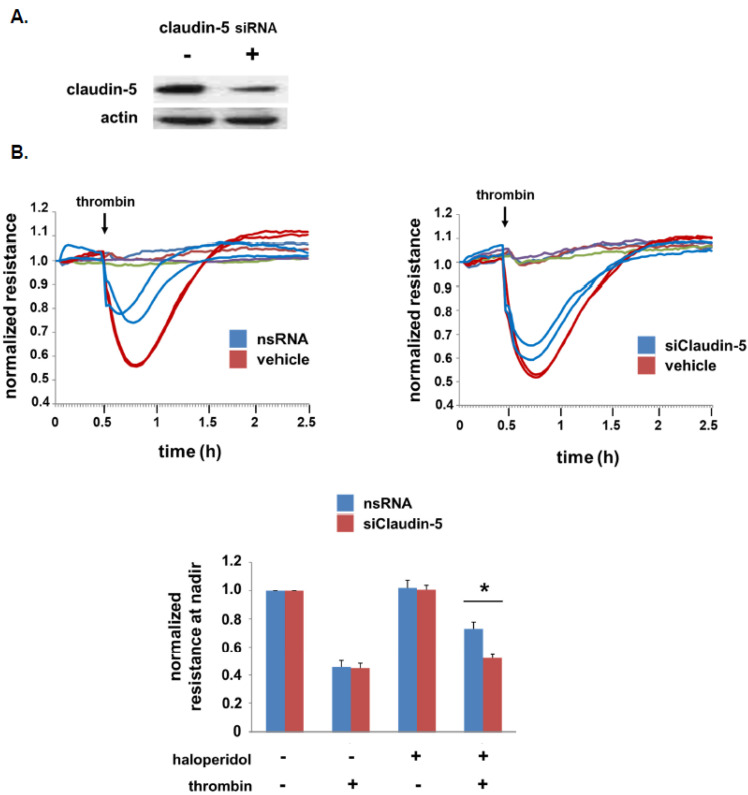
EC barrier protection by haloperidol is mediated by claudin-5. Human pulmonary artery ECs were grown to confluence overlying gold-plated microelectrodes for measurements of transendothelial electrical resistance. Cells were transfected with siRNA specific for claudin-5 (siClaudin-5) or non-specific siRNA (nsRNA) for 3 days with silencing confirmed by Western blotting (**A**). Cells were then treated with haloperidol (10 µM) for 24 h followed by thrombin stimulation (1 U/mL). Real-time measurements of normalized resistance were recorded and the nadir for each condition was used for the bar graph (**B**) (representative tracings shown; *n* = 3/condition, * *p* < 0.05).

**Figure 4 cells-10-02186-f004:**
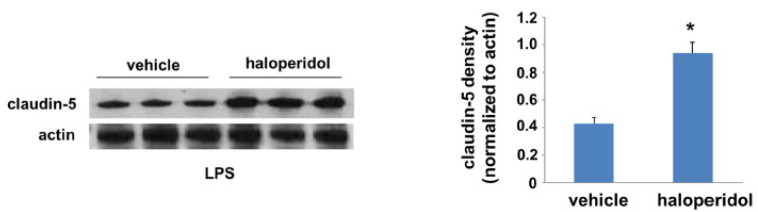
Lung claudin-5 expression is increased by haloperidol in vivo. A murine model of ALI was utilized to study the effects of haloperidol in vivo. Lungs were harvested from LPS-treated animals that had been pretreated with haloperidol (4 mg/kg, IP) or vehicle for 24 h and whole lung homogenates were used for Western blotting of claudin-5 (representative blots shown; *n* = 3/group, * *p* < 0.05 compared to controls).

**Figure 5 cells-10-02186-f005:**
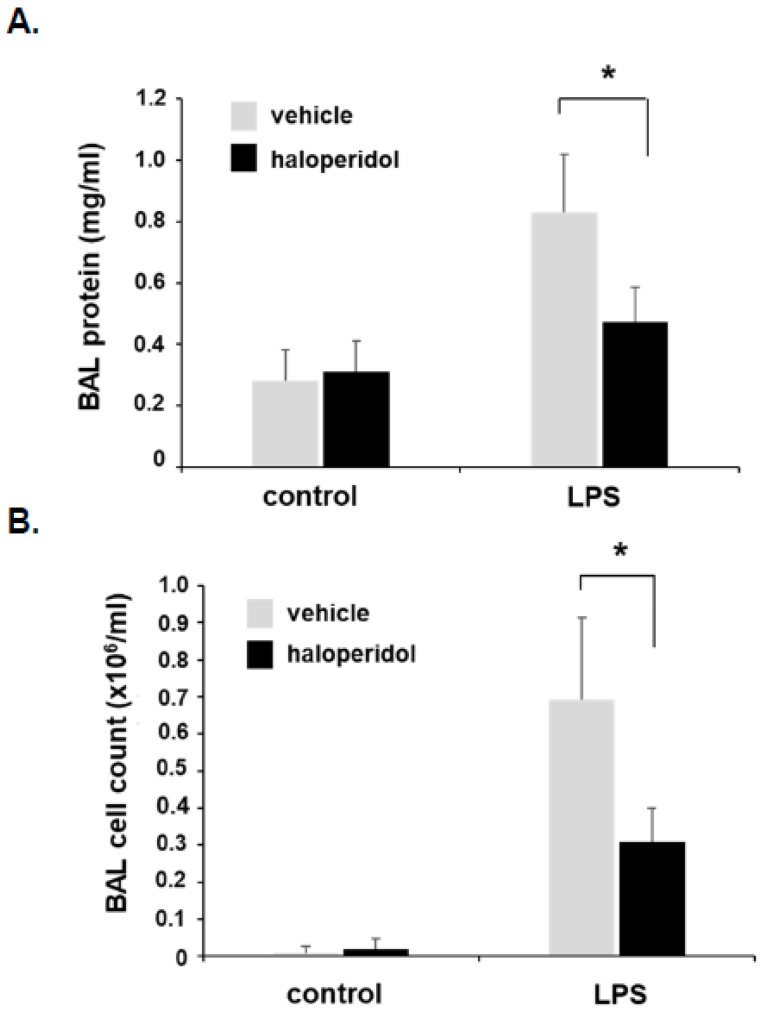
Murine ALI is attenuated by haloperidol: BAL fluid protein and cell counts. Mice were pretreated with haloperidol (4 mg/kg, IP) or vehicle 24 h prior to repeat treatment of the same, followed by the administration of intratracheal LPS (1.25 mg/kg). BAL fluid was collected 24 h later and assessed for (**A**) total protein content and (**B**) total cell counts (*n* = 5/LPS groups, * *p* < 0.05).

**Figure 6 cells-10-02186-f006:**
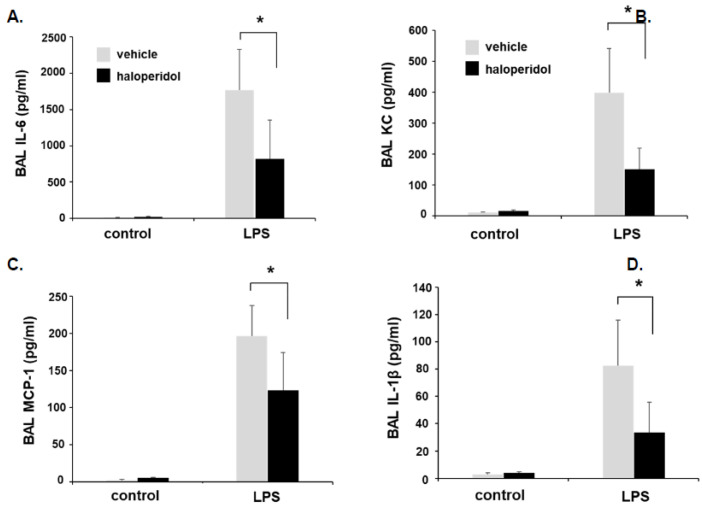
Murine ALI is attenuated by haloperidol: BAL fluid cytokines. Mice were pretreated with haloperidol (4 mg/kg, IP) or vehicle 24 h prior to repeat treatment of the same followed by the administration of intratracheal LPS (1.25 mg/kg). BAL fluid was collected 24 h later and assessed for inflammatory cytokines including (**A**) IL-6, (**B**) KC, (**C**) MCP-1, and (**D**) IL-1β. (*n* = 5/LPS groups, * *p* < 0.05).

**Figure 7 cells-10-02186-f007:**
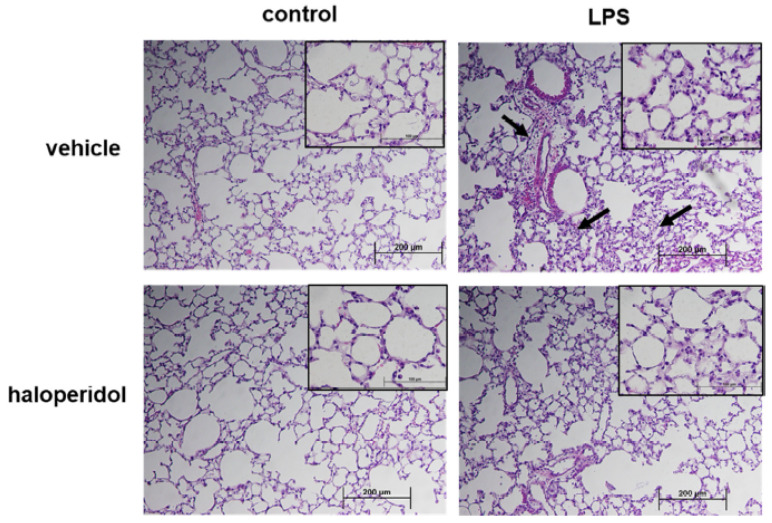
Murine ALI is attenuated by haloperidol: lung histology. Mice were pretreated with haloperidol (4 mg/kg, IP) or vehicle 24 h prior to repeat treatment of the same followed by the administration of intratracheal LPS (1.25 mg/kg). Lungs were harvested and utilized for histology evaluation. LPS-induced inflammatory cell infiltration and interstitial edema (arrows) was less evident in animals that were pre-treated with haloperidol (representative 10× images shown with 40× inset).

## Data Availability

The data presented in this study are available on request from the corresponding author.

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
