# Peer review of "Haloperidol Attenuates Lung Endothelial Cell Permeability In Vitro and In Vivo"

_cells, 2021, doi:10.3390/cells10092186_

Round 1

Reviewer 1 Report

Reviewing_Cells_Colamonici_2021

The submitted manuscript by Colamonici and colleagues in the journal shows that haloperidol and lithium chloride, two antipsychotic medications, have similar lung vascular-protective effects via an upregulation of 3 of the tight junctions, the Claudin-5, in addition to occludens and ZO-1 expression. The manuscript is well written with a logical progression. 

Even if the data regarding the effects of the cell permeability are interesting, the authors overinterpret their results. Indeed even if the use of haloperidol and lithium chloride is associated with an upregulation of the protein levels of claudin-5, occludens and ZO-1, the authors could not conclude of a specific mechanism but just of an association. All the more that the authors do not report any data about RNA levels of the tight junctions. Furthermore the authors report data about haloperidol and lithium in the in vitro model but only about haloperidol in the murine model. In order to improve the manuscript and improve the understanding of the readers of the journal, the paper should be uniformize either reporting results about haloperidol and lithium (so it is required to add experiments) or to focus only on haloperidol.

Major points 

Some elements such as the doses or the exposition time are reported in the results. In order to improve the understanding of the readers, the authors should group together all the information in the “Material and methods”. 

Why did the authors report results with 3 different doses of haloperidol and lithium in the experiments related to figure 1 and figure 2 and then only 1 dose for the other experiments ? If the authors decided the dose of 10µM because they found the bigger effect on the protein expressions at this dose, they should explain this point in the methods.  

Why did not the authors test lithium chloride in their murine model ? Even if the results are attractive the authors have to report results using lithium chloride in a murine model. Or the authors should remove results of in vitro experiments with lithium chloride and focus the manuscript on the effects of haloperidol. Indeed it is not logical to report results of in vitro and in vivo models with not the same reactives.  

Concentrations of haloperidol (4mg/kg) used in the murine model are very high by comparison of the doses used in patients (usually around 2.5-5mg three times a day for an adult weighing 70 kg). Could the authors discuss this point ?

Line 167 : The sentence should be in the discussion instead of the results. 

A major limitation is that the author only reported expression at the protein level and not data are reported of the RNA for the three tight junctions. Could the authors report such a result ? If the can’t, the authors should add this point as a limitation in the discussion

The author should moderate the discussion avoiding the term of “upregulation” and mostly use “association” or equivalent words. 

The author should add data about the safety (or not) to use lithium chloride in ICU. Indeed haloperidol and lithium chloride are 2 medications with potential fatal adverse events, especially if they are used for a long time, all the more if they are used in specific situations such as in ICU. 

Minor points

Line 86: Standardize the font size for “0.1 M Sodium Orthovanadate”

Line 93: Standardize the font size for “http://imagej.nih.gov/ij/”.

Line 113: tip error “Haloperidol” instead of “haloprerdol”

Line 236 : Remove space ICU

Reviewer 2 Report

In this study, Marco Colamonici and colleagues evaluated the role of haloperidol and lithium chloride in regulating lung vascular permeability. In vitro studies showed that pre-treatment of lung artery endothelial cells with haloperidol or lithium chloride attenuated thrombin-induced barrier dysfunction, which might be associated with upregulation of claudin-5 or other junctional proteins (e.g. occludin and ZO-1). In vivo, mice pretreated with haloperidol exhibited lower lung inflammation and injury induced by LPS. Overall, this is a well written manuscript, the experimental design is reasonable, and the results are significant. However, the mechanism investigation is missing.

Major concerns.

  1. Haloperidol or lithium chloride has multiple biological effects, as mentioned in the discussion. To investigate whether claudin-5 mediates their protective effects in barrier function, silencing or blocking claudin-5 experiments are warranted (at least for in vitro experiments).
  2. Hyperpermeability mostly occurs in microvascular vessels. What is the rationale of using artery, not microvascular, endothelial cells?
  3. Although the results showed statistical difference, the n numbers are too low. Suggest to increase n≥5.
  4. In vivo measurement of claudin-5, occlduin, and ZO-1 is required.
  5. Why was the in vivo effect of lithium chloride not evaluated?

Round 2

Reviewer 1 Report

The authors have strongly improved the

manuscrit and i havn’t any concern.

Congratulations 

Thanks. 

All the best 

Reviewer 2 Report

This authors have appropriately addressed my concerns. I have no more comments. I believe the manuscript has been sufficiently improved to warrant publication in Cells.